# Sum Estimation under Personalized Local Differential Privacy

**Dajun Sun**[1]    **Wei Dong**[2*]    **Yuan Qiu**[3*]    **Ke Yi**[1]    **Graham Cormode**[4]

[1]Hong Kong University of Science and Technology    [2]Nanyang Technological University
[3]Southeast University    [4]University of Warwick
dsunad@connect.ust.hk   wei_dong@ntu.edu.sg   yuanqiu@seu.edu.cn
yike@cse.ust.hk   g.cormode@warwick.ac.uk

## Abstract

People have diverse privacy requirements. This is best modeled using a personalized local differential privacy model where each user privatizes their data using a possibly different privacy parameter. While the model of personalized local differential privacy is a natural and important one, prior work has failed to give meaningful error bounds. In this paper, we study the foundational sum/mean estimation problem under this model. We present two novel protocols that achieve strong error guarantees. The first gives a guarantee based on the radius of the data, suiting inputs that are centered around zero. The second extends the guarantee to the diameter of the data, capturing the case when the points are situated arbitrarily. Experimental results on both synthetic and real data show that our protocols significantly outperform existing methods in terms of accuracy while providing a strong level of privacy.

## 1 Introduction

Sum/mean estimation under differential privacy (DP) is a fundamental building block in privacy-preserving machine learning [1, 5], statistical analysis [14], and query processing [19, 22]. Among the various models of DP, the local model (LDP) has attracted much attention, as it makes no trust assumptions and is easy to implement in a distributed setting. In this model, each user privatizes their own data, usually by adding some noise, and sends the noisy result to an untrusted analyst. However, existing work on LDP assumes that all users adopt the same privacy parameter ($\varepsilon$ or $\rho$) when privatizing their data, which is an overly simplistic assumption. In practice, people have diverse privacy requirements: Conservative users might be unwilling to share data, while more liberal ones are happy to contribute. Indeed, many apps give users the option of sharing or not sharing their data, which can be considered the most coarse-grained personalized privacy.

In this paper, we consider a fine-grained and more quantitative personalized LDP model where each user $u$ is allowed to set their privacy parameter $\Phi(u)$ to any positive real number, with smaller values corresponding to higher privacy requirement. In particular, we adopt *zero-concentrated DP*, where $\Phi(u)$ corresponds to the parameter $\rho$ (the formal definition is given in Section 3). We study the sum estimation problem under such a setting: Let $\mathcal{U} = \{u_1, ..., u_n\}$ be the set of users. We model the given instance as a function $\mathbf{I} : \mathcal{U} \to [B]^d$, where $[B] := \{0, \ldots, B\}$ and $\|\mathbf{I}(u)\|_2 \le B$ for all $u$, i.e., each user $u$ holds a $d$-dimensional non-negative integer-valued vector $\mathbf{I}(u)$ in a ball of radius $B$. This is without much loss of generality: real-valued vectors can be translated, scaled, and rounded with negligible precision loss as long as $B$ is sufficiently large, say, $2^{32}$. In this paper, we focus on estimating the core function of $\mathrm{Sum}(\mathbf{I}) = \sum_{u \in \mathcal{U}} \mathbf{I}(u)$; mean estimation results follow easily.

---

*Corresponding authors.

39th Conference on Neural Information Processing Systems (NeurIPS 2025).

In an LDP protocol, each user privatizes their own data by themselves using a local randomizer, so it automatically translates to this personalized setting. For instance, each user can apply the Gaussian mechanism to $\mathbf{I}(u)$, which adds a Gaussian noise with scale $O\left(\frac{B}{\sqrt{\Phi(u)}}\right)$ to each coordinate [10, 38, 35].

However, this simple solution may fail miserably. First, as $B$ an *a priori* upper bound on the norm, it must be set conservatively large, resulting in excessive noise. Meanwhile, the norms of data in typical instances are much smaller than $B$, so some instance-specific error, e.g., one that is proportional to the *radius* $\mathrm{rad}(\mathbf{I}) = \max_{u\in\mathcal{U}} \|\mathbf{I}(u)\|_2$ or the *diameter* $\omega(\mathbf{I}) = \max_{u_i, u_j \in \mathcal{U}} \|\mathbf{I}(u_i) - \mathbf{I}(u_j)\|_2$, would be more desirable. Second, this solution is susceptible to highly conservative users with very small $\Phi(u)$, who may just as well be removed from the analysis.

An effective approach to addressing both issues is to truncate/clip the data before adding noise. Given a threshold $\tau$, we truncate the data on the $\ell_2$ norm, i.e., set $\bar{\mathbf{I}}(u, \tau) = \min(\|\mathbf{I}(u)\|_2, \tau)\frac{\mathbf{I}(u)}{\|\mathbf{I}(u)\|_2}$. Then adding a Gaussian noise with scale $O\left(\frac{\tau}{\sqrt{\Phi(u)}}\right)$ satisfies the LDP requirement for $u$. However, it is not clear how to select a good $\tau$ in the personalized LDP model. More critically, even if a best $\tau$ could be found, applying the same $\tau$ to all users would still yield sub-optimal results. Consider the following 1D example.

**Example 1.1.** Suppose user $i$ has (scalar) data $\mathbf{I}(u_i) = i$ with privacy parameter $\Phi(u_i) = (n+1-i)^2/n$, for $i = 1, \ldots, n$, representing a typical case where users with larger data values also have stronger privacy requirements. Directly applying the Gaussian mechanism corresponds to $\tau = B$, and incurs a total error of $O\left(B\sqrt{\sum_i (1/\Phi(u_i))}\right) = O(B\sqrt{n})$. More generally, the truncation mechanism above returns $\min(\mathbf{I}(u), \tau) + \mathcal{N}(0, \frac{\tau^2}{\Phi(u)})$. Using the notation $(x)^+ := \max(x, 0)$, the total error is

$$\sum_{i=1}^{n}(\mathbf{I}(u_i) - \tau)^+ + \sqrt{\sum_{i=1}^{n} \frac{\tau^2}{\Phi(u_i)}}, \tag{1}$$

where the first term is the truncation bias and the second is the total error of the noise. Even with the optimal $\tau = n - \sqrt{n}$ (which is not clear how to obtain without knowledge of the users' private data), both the bias and the noise term are $O(n^{3/2})$. $\square$

In Section 4, we present a protocol that achieves an $\ell_2$ error of[1]

$$\min_{s\in\mathbb{R}_{\geq 0}} \left( \left\| \sum_u \left( \|\mathbf{I}(u)\|_2 - s\sqrt{\Phi(u)} \right)^+ \frac{\mathbf{I}(u)}{\|\mathbf{I}(u)\|_2} \right\|_2 + \tilde{O}(s\sqrt{nd}) \right) \tag{2}$$

in $d$ dimensions. It has a similar form to (1), but with two key differences: First, instead of a uniform truncation threshold $\tau$ for all users $u$, we make it proportional to $\sqrt{\Phi(u)}$. Intuitively, this allows us to obtain more information about more liberal users. It truncates more aggressively on conservative users (such as $u_n$ in Example 1.1), but this also reduces the noise they introduce to the final estimate. Second, our protocol automatically selects the optimal scaling factor $s$, in one round and in a DP fashion. These two improvements allow us to reduce the error significantly. When applied to the instance in Example 1.1, (2) is $\tilde{O}(n^{4/3})$, achieved by $s = n^{5/6}$. Also, note that in the uniform-privacy LDP setting where $\Phi(\cdot) \equiv \rho$, (2) degenerates into $\tilde{O}\left(\mathrm{rad}(\mathbf{I})\sqrt{nd/\rho}\right)$, achieved by $s = \|\mathbf{I}\|_2^{(\sqrt{nd/\rho})}/\sqrt{\rho}$, where $\|\mathbf{I}\|_2^{(k)}$ denotes the $k$-th largest $\ell_2$ norm in $\mathbf{I}$. This matches the radius-dependent bound of the LDP protocol in [18]. In addition to the $\ell_2$ error, our protocol also achieves a similar error guarantees in terms of $\ell_\infty$, which allows us to solve some related problems like frequency estimation, range counting, and quantiles in the personalized LDP model.

**Example 1.2.** Next, consider a variant of Example 1.1 where the users' data are clustered away from the origin: $\mathbf{I}(u_i) = B/2 + i$ for $i = 1, \ldots, n$ (assuming $B \gg n$). On this instance, the error bound (2) becomes $\tilde{O}(B\sqrt{n})$, no better than the naive Gaussian mechanism. $\square$

In Section 5 we present another protocol that achieves the following diameter-dependent error bound:

$$\tilde{O}\left( \min_{s\in\mathbb{R}_{\geq 0}} \left( \omega(\mathbf{I})\sqrt{\sum_u \mathbb{1}\left(s\sqrt{\Phi(u)/2} < \omega(\mathbf{I})\right)} + s\sqrt{nd} \right) \right). \tag{3}$$

Since $\omega(\mathbf{I}) \leq 2 \cdot \mathrm{rad}(\mathbf{I})$ on any instance $\mathbf{I}$, while $\omega(\mathbf{I})$ could much smaller than $\mathrm{rad}(\mathbf{I})$, such a diameter-dependent bound is more preferable, especially for datasets that are clustered away from

---

[1]The $\tilde{O}$ notation hides logarithmic factors.

the origin. For the instance in Example 1.2, we have $\omega(\mathbf{I}) = n$ while $\mathrm{rad}(\mathbf{I}) = B + n$, and (3) is $\tilde{O}(n^{4/3})$, matching what we can achieve on the instance in Example 1.1. Furthermore, in the uniform-privacy LDP setting, (3) degenerates into $\tilde{O}\big(\omega(\mathbf{I})\sqrt{nd/\rho}\big)$, achieved by $s = \omega(\mathbf{I})\sqrt{2/\rho}$, matching the diameter-dependent bound of the LDP protocol in [18].

## 2  Related Work

Providing personalized privacy protection is a well motivated problem due to the diversity of users. In fact, this issue was studied even before DP became the primary privacy model. For example, Xiao and Tao [34] defined a notion of personalized privacy in the context of $k$-anonymity, which was later extended to other related privacy models [37, 17, 30]. However, these models do not provide privacy protection as rigorous as differential privacy [26, 39].

The model of personalized differential privacy (also known as heterogeneous differential privacy) was initialized under the central model of DP by Jorgensen et al. [20], where a trusted central data curator holds and analyzes users' data. They designed a general-propose sampling mechanism and extended the inverse sensitivity-based exponential mechanism [27, 11, 6] to PDP, but without formal guarantees on the utility. Recent work by Sun et al. [31] provides the first result on sum estimation and private query answering with rigorous utility guarantees. Some baseline ML problems such as support vector machine and linear regression have been studied in [24]. Additionally, [15] studies how to track the privacy consumption of each user over multiple queries.

Personalized differential privacy under the local model has also been studied for federated learning [25, 36], point-of-interest recommendation [7], mean estimation [35], and statistical histograms [38]. However, these works simply use the naive Gaussian mechanism that adds a noise with scale $\frac{B}{\sqrt{\Phi(u)}}$ for each $\mathbf{I}(u)$. For this to succeed, they have to use a small $B$ and assume that the norms of all data are not much smaller than $B$.

There are also some other interesting papers that do not study the PDP model directly, but nevertheless have a "personalized" flavor. For example, [33, 28] consider a personalized privacy setting where each user determines which part of its data is public or private, and then provide (standard) DP protection only on the private part. Zhang et al. [40] points to another interesting direction called multi-analyst DP. Recent work by Seeman et al. [29] studies the notion called per-record DP, where each user has a different privacy level depending on the content of his record.

## 3  Preliminaries

We consider the local model of differential privacy where each user $u$ retains their data $\mathbf{I}(u)$, and only sends $\mathcal{M}(\mathbf{I}(u))$ to the *analyzer*, where $\mathcal{M}(\cdot)$ is called a *local randomizer*, which must satisfy (local) DP. Each user $u$ has a possibly different privacy parameter $\Phi(u)$, where $\Phi : \mathcal{U} \to \mathbb{R}^+$ is called the *privacy specification*, known to the analyzer. Define $\rho_{\min} := \min_{u \in \mathcal{U}} \Phi(u)$ and $\rho_{\max} := \max_{u \in \mathcal{U}} \Phi(u)$.

There are several versions of DP. In this paper, we adopt *zero-Concentrated Differential Privacy* (CDP) [9], which is more suitable for high-dimensional data. It naturally fits the personalized setting:

**Definition 3.1** (Personalized local zCDP (PLCDP))**.** *For a given privacy specification $\Phi$, a local randomizer $\mathcal{M}$ satisfies $\Phi$-PLCDP if for any user $u$, any $\mathbf{I}(u), \mathbf{I}'(u)$, and any $\alpha > 1$,*

$$D_\alpha\left(\mathcal{M}(\mathbf{I}(u))\|\mathcal{M}(\mathbf{I}'(u))\right) \le \alpha \cdot \Phi(u),$$

*where $D_\alpha(\cdot\|\cdot)$ denotes the $\alpha$-Rényi divergence between the distributions of the two random variables.*

It is known that the privacy guarantee provided by CDP is sandwiched between that of pure DP and approximate DP, with their parameters roughly related as $\varepsilon = \tilde{\Theta}(\sqrt{\rho})$ [9]. The canonical mechanism for achieving PLCDP adds Gaussian noise with proper scale to each coordinate of the data:

**Lemma 3.1** (Gaussian Mechanism [9])**.** *Under the constraint that $\|\mathbf{I}(u)\|_2 \le B$ for all $u$, the randomizer $\mathcal{M}$ that outputs $\mathcal{M}(\mathbf{I}(u)) = \mathbf{I}(u) + \mathcal{N}\left(0, \frac{B^2}{2\Phi(u)} \cdot \mathbf{1}_{d \times d}\right)$ satisfies $\Phi$-PLCDP.*

In a one-round protocol, the analyzer $\mathcal{A}$ collects $\mathcal{M}(\mathbf{I}(u))$ for all $u \in \mathcal{U}$ and outputs $\mathcal{A}((\mathcal{M}(\mathbf{I}(u)))_u)$. It is also possible for a protocol to run over multiple rounds, in which case the privacy consumption

accumulates. For simplicity, we only state and prove (in Appendix A.1) a 2-round version; extension to more rounds is straightforward.

**Lemma 3.2** (Adaptive composition). *Let $\mathcal{M}_1(\cdot)$ and $\mathcal{M}_2(\cdot, y)$ be local randomizers such that $\mathcal{M}_1(\cdot)$ satisfies $\Phi_1$-PLCDP and $\mathcal{M}_2(\cdot, y)$ satisfies $\Phi_2$-PLCDP for any $y$. Then the 2-round protocol that collects $(\mathcal{M}_1(\mathbf{I}(u)))_u$ in the first round and $(\mathcal{M}_2(\mathbf{I}(u), y(u)))_u$ in the second round satisfies $(\Phi_1 + \Phi_2)$-PLCDP, where $y(u)$ may depend on $(\mathcal{M}_1(\mathbf{I}(u)))_u$.*

Note that if $\mathcal{M}_2(\cdot, y)$ does not depend on $y$, then the composition is non-adaptive, and the two randomizers can be run in the same round.

We also need the following tail bound of the Gaussian distribution for utility analysis:

**Lemma 3.3** (Gaussian tail). *If $X \sim \mathcal{N}(0, \sigma^2)$, then $\Pr\left[|X| > \sigma\sqrt{2\ln\frac{2}{\beta}}\right] \leq \beta$ for any $0 < \beta < 1$.*

# 4 Radius-Dependent Protocol

In this section, we present a one-round PLCDP protocol that achieves the following error guarantee:

**Theorem 4.1.** *For any $\Phi$, the local randomizer defined in Algorithm 1 satisfies $\Phi$-PLCDP. For any $\beta$, the analyzer can run Algorithm 2 to obtain an estimate of $\mathrm{Sum}(\mathbf{I})$ with an $\ell_2$ error at most*

$$\min_{s \in \mathbb{R}_{\geq 0}} \left( \left\| \sum_u \left( \|\mathbf{I}(u)\|_2 - s\sqrt{2\Phi(u)} \right)^+ \frac{\mathbf{I}(u)}{\|\mathbf{I}(u)\|_2} \right\|_2 + 4s\sqrt{2ndt\ln\frac{2td}{\beta}} \right) \tag{4}$$

*and an $\ell_\infty$ error at most*

$$\min_{s \in \mathbb{R}_{\geq 0}} \left( \left\| \sum_u \left( \|\mathbf{I}(u)\|_2 - s\sqrt{2\Phi(u)} \right)^+ \frac{\mathbf{I}(u)}{\|\mathbf{I}(u)\|_2} \right\|_\infty + 4s\sqrt{2nt\ln\frac{2td}{\beta}} \right) \tag{5}$$

*with probability at least $1 - \beta$, where $t = \log\left(B\sqrt{\frac{\rho_{\max}}{\rho_{\min}}}\right)$.*

Below, we describe our randomizer and analyzer, while giving some intuition why they can achieve the error bounds in Theorem 4.1, with the formal proof in Appendix A.2.

Our local randomizer invokes the truncation mechanism on user $u$ with a truncation threshold $\tau(u) = s\sqrt{2\Phi(u)}$. In order to find an optimal $s$ up to a constant factor, we try a logarithmic number of possible values from $s_{\min} = \frac{1}{\sqrt{2\rho_{\max}}}$ to $s_{\max} = \frac{B}{\sqrt{2\rho_{\min}}}$. More precisely, letting $t = \log\frac{s_{\max}}{s_{\min}} = \log\left(B\sqrt{\frac{\rho_{\max}}{\rho_{\min}}}\right)$, we try $s_i = 2^i \cdot s_{\min}$ for $i = 0, 1, ..., t$, i.e., invoke $t$ instances of the truncation mechanism concurrently with $\tau_i(u) = 2^i\sqrt{\frac{\Phi(u)}{\rho_{\max}}}$, while splitting the privacy budget using the non-adaptive version of Lemma 3.2. The details are given in Algorithm 1.

After receiving the $t$ noisy truncated vectors from all users, the analyzer adds them up respectively. It remains for the analyzer to pick one out of the $t$ noisy truncated sums that achieves a near-optimal error. For this, we use a "subtract-max" technique [13]: For each dimension, we subtract a term proportional to the noise scale from each noisy sum and take the maximum. The details are given in Algorithm 2. The intuition that this can find an optimal $s$ is as follows. We know that that the optimal $s$ should balance the bias and noise. A small $s$ introduces a large bias but small noise, so it is unlikely to be the maximum. A large $s$ has small bias but large noise, so subtracting a term proportional to the noise scale turns it into an underestimate. Between these two extremes, the underestimate where the bias matches the noise has the best chance to become the maximum.

**Remark.** As stated, the randomizer in Algorithm 1 sends out a message of size $\tilde{O}(d)$. Using the lossless compression technique in [16], this can be compressed to $\tilde{O}(1)$ with negligible loss in the privacy and utility. On the other hand, instead of reporting a value for all $s_i$, each user may sample only one scale and send the corresponding truncated value. This will not affect the asymptotic accuracy of the algorithm but can reduce communication by a factor of $\log\sqrt{\frac{\rho_{\max}}{\rho_{\min}}}B$.

Below, we discuss two applications of our radius-dependent protocol, which will also be useful in our diameter-dependent protocol in Section 5.

---

**Algorithm 1:** LocalSum-R (Randomizer)

---

**Input:** $\mathbf{I}(u)$, $\Phi$, $B$, $d$

1   $t \leftarrow \log\left(\sqrt{\frac{\rho_{\max}}{\rho_{\min}}}B\right)$;

2   **for** $i \leftarrow 0, 1, \ldots, t$ **do**

3     Let $s_i \leftarrow \frac{2^i}{\sqrt{2}\rho_{\max}}, \tau_i(u) \leftarrow s_i\sqrt{2\Phi(u)}$;

4     Define $\mathbf{I}_i(u) = \min(\|\mathbf{I}(u)\|_2, \tau_i(u))\frac{\mathbf{I}(u)}{\|\mathbf{I}(u)\|_2}$;

5     $\widetilde{\mathbf{I}}_i(u) \leftarrow \mathbf{I}_i(u) + \mathcal{N}\left(0, s_i^2 \cdot t \cdot \mathbf{1}_{d \times d}\right)$;

6   **end**

7   **return** $\{\widetilde{\mathbf{I}}_i(u)\}_{i=0}^t$;

---

---

**Algorithm 2:** LocalSum-A (Analyzer)

---

**Input:** $\{\widetilde{\mathbf{I}}_i\}_{i=0}^t$, $\Phi$, $B$, $\beta$, $d$

1   $t \leftarrow \log\left(\sqrt{\frac{\rho_{\max}}{\rho_{\min}}}B\right)$;

2   **for** $i \leftarrow 0, 1, \ldots, t$ **do**

3     $s_i \leftarrow \frac{2^i}{\sqrt{2}\rho_{\max}}$;

4     $\widetilde{\text{Sum}}_i(\mathbf{I}) \leftarrow \text{Sum}(\widetilde{\mathbf{I}}_i) - s_i\sqrt{2nt\ln\frac{2td}{\beta}} \cdot \mathbf{1}$;

5   **end**

6   **return** $\widetilde{\text{Sum}}(\mathbf{I})$ such that $\widetilde{\text{Sum}}(\mathbf{I})[j] \leftarrow \max\{\max_i \widetilde{\text{Sum}}_i(\mathbf{I})[j], 0\}$, where $\widetilde{\text{Sum}}_i(\mathbf{I})[j]$ is the $j$th coordinate of $\widetilde{\text{Sum}}_i(\mathbf{I})$;

---

**Histogram (frequency estimation).** In the histogram problem, each user holds an element $\mathbf{I}(u) \in [B]$, and the goal is to obtain a private histogram from which we can estimate the number of occurrences of any $i \in [B]$. By taking $\mathbf{I}(u)$ as a one-hot vector in $B$ dimensions, the histogram problem becomes a sum estimation problem, and the $\ell_\infty$ error bound (5) provides a guarantee on the maximum error on the estimated frequency of any $i \in [B]$. For this special case, (5) can be further simplified as:

**Corollary 4.2.** *Given $\Phi$, $\beta$, $B$, $\mathbf{I}(u) \in [B]$, Algorithms 1 and 2 return a private histogram such that for all $i \in [B]$, the frequency of $i$ can be estimated with error at most*

$$O\left(k\sqrt{n\log\frac{\rho_{\max}}{\rho_{\min}}\log\frac{\log B \log\frac{\rho_{\max}}{\rho_{\min}}}{\beta}}\right),$$

*where $k$ is the smallest index such that $\sum_{i=1}^k \sqrt{\Phi(u_i)} \geq 1$, assuming the users are ranked in the non-decreasing order of $\Phi(u_i)$.*

Note that when $\Phi_i(u) \equiv \rho$, we have $k = 1/\sqrt{\rho}$, and the error degenerates to $\tilde{O}(\sqrt{n/\rho})$, matching the error bound in the standard LDP model [23].

**Range counting and quantiles.** The range counting problem has the same setup as above, but we are interested in counting the number of elements in any range $[L, R] \subseteq [B]$. Note that the histogram problem is the special case where $L = R$. The range counting problem can be reduced to $\log B$ instances of the histogram problem by decomposing the universe $[B]$ in a hierarchical fashion. The following theorem summarizes the result, with details given in Appendix A.3, A.4.

**Theorem 4.3.** *Given $\Phi$, $\beta$, $B$, $\mathbf{I}(u) \in [B]$, with probability at least $1 - \beta$, all range counting queries over $[B]$ can be answered with error*

$$O\left(k\sqrt{n\log\frac{\rho_{\max}}{\rho_{\min}}\log\frac{\log B \log\frac{\rho_{\max}}{\rho_{\min}}}{\beta}\log^2 B}\right) \tag{6}$$

*under $\Phi$-PLCDP, where $k$ is as defined in Corollary 4.2.*

Finally, using range counting queries in the form of $[1, x]$, we can do a binary search on $[B]$ to find any quantile (e.g., the median) approximately. Then (6) becomes the *rank error* of the returned quantile (e.g., the returned median is ranked at $\frac{n}{2} \pm$ (6)).

## 5 Diameter-Dependent Protocol

The protocol above achieves an error that scales with the radius, i.e., the maximum $\ell_2$ norm, which only works well for datasets that are around the origin. For the general case, it is more desirable to achieve an error that scales with the diameter $\omega(\mathbf{I})$ rather than $\mathrm{rad}(\mathbf{I})$. In this section, we design such a PLCDP protocol, although it requires two rounds.

Our solution is to first shift the dataset towards the origin such that the *radius* of the shifted dataset is roughly the *diameter* of the original dataset. That is, the shifted dataset should be concentrated around the origin, and it preserves the diameter of the original dataset. Then we can apply the previous (radius) sum algorithm to the shifted dataset and shift the result back.

To achieve this goal, we need to find an interior point (in our case, the median) on each dimension independently, using the PLCDP quantile selection algorithm described above. Since this is done on all the $d$ dimensions, privacy parameters need to be further divided into $d$ parts. Below is the guarantee that follows directly from Theorem 4.3:

**Corollary 5.1.** *Given* $\beta, \Phi, \mathbf{I}(u) \in [B]^d$, *if* $n \geq cd \log \frac{\rho_{\max}}{\rho_{\min}} \log \frac{d \log B \log \frac{\rho_{\max}}{\rho_{\min}}}{\beta} k^2 \log^4 B$ *for some constant* $c$, *with probability at least* $1 - \beta$, *we can find an interior point in each dimension while preserving* $\Phi$-*PLCDP.*

However, in high dimensions, doing such a shift in each dimension may 'expand' the dataset and result in a radius of $O(\sqrt{d}\omega(\mathbf{I}))$. Consider the following example.

**Example 5.1.** Consider a dataset consisting $d$ unit vectors in $d$-dimensional space, the diameter is $\sqrt{2}$. Obviously, in every single dimension 1 is an interior point, but shifting with $(1, 1, ..., 1)$ results in a radius of $\sqrt{d-1}$.

The reason is that the values in each dimension may be skewed. In order to preserve the diameter of $\mathbf{I}$ in high dimensions, we perform a random rotation to 'balance' the values before estimating the median. The rotation is done by $\hat{\mathbf{I}}(u) := HD\mathbf{I}(u)$, where $H$ is the $d \times d$ Hadamard matrix, and $D$ is a $d \times d$ diagonal matrix whose diagonal entry is independently and uniformly drawn from $\{-1, +1\}$. This process can be done via public randomness and does not need additional communication. The following Lemma [3] says the rotated data is likely to be more 'balanced':

**Lemma 5.2** ([3]). *Let* $H$ *and* $D$ *be defined as above. Then, for any* $\boldsymbol{x} \in \mathbb{R}^d$ *and any* $\beta > 0$,

$$\Pr \left[ \|HD\boldsymbol{x}\|_\infty \geq \|\boldsymbol{x}\|_2 \cdot \sqrt{2 \log \frac{4d}{\beta}} \right] \leq \beta.$$

For any pair of users $u_1, u_2$, by setting $\boldsymbol{x} = \mathbf{I}(u_1) - \mathbf{I}(u_2)$, the above lemma says with high probability, the maximum distance between their rotated data on each dimension $\left\| \hat{\mathbf{I}}(u_1) - \hat{\mathbf{I}}(u_2) \right\|_\infty$ is no greater than $\sqrt{2 \log \frac{4d}{\beta}} \|\mathbf{I}(u_1) - \mathbf{I}(u_2)\|_2$. To make this hold simultaneously for all pairs of points, we apply a union bound and the distance bound will become $O(\sqrt{\log \frac{nd}{\beta}} \omega(\mathbf{I}))$, thus the radius of the shifted data $\hat{\mathbf{I}}_s$ is $\tilde{O}(\sqrt{d}\omega(\mathbf{I}))$. After estimating the sum of the rotated data, we should rotate this result back by multiplying $(HD)^{-1}$, which decreases the $\ell_2$ norm by a factor of $\frac{1}{\sqrt{d}}$ so the additional $\sqrt{d}$ factor here will be eliminated finally.

The whole process of sum estimation can be formulated as follows:

1. Each user does a random rotation on their data using public information $H, D$, denote the rotated data as $\hat{\mathbf{I}}(u)$.

2. Apply the median selection protocol on each dimension using the technique described in Section 4 with privacy $\frac{\Phi}{2d}$ and failure probability $\frac{\beta}{4d}$. Note for original data such that $\|\mathbf{I}(u)\|_2 \leq B$, the rotated data will have $\|\hat{\mathbf{I}}(u)\|_2 \leq dB$ and its coordinate will lie in $[dB]$.

3. Analyzer receives messages from users and returns the estimated median of each dimension; denote the median vector as $\widetilde{\boldsymbol{m}} \in [dB]^d$ where the $j$th coordinate $\widetilde{\boldsymbol{m}}[j]$ represents the estimated median of dimension $j$.

4. Each user shifts the rotated data towards the median and splits the shifted data into negative/positive parts. Let the shifted data be $\hat{\mathbf{I}}_s(u) = \hat{\mathbf{I}}(u) - \widetilde{\boldsymbol{m}}$ and define the negative part as $\hat{\mathbf{I}}_s^-(u) = -\min(\hat{\mathbf{I}}_s(u), 0)$ (on each coordinate) whereas the positive part is $\hat{\mathbf{I}}_s^+(u) = \max(\hat{\mathbf{I}}_s(u), 0)$ (on each coordinate). This is to ensure each part contains only non-negative values as required by our sum estimation protocol.

5. Each user applies the sum protocol described in Section 4 on $\hat{\mathbf{I}}_s^-$ and $\hat{\mathbf{I}}_s^+$ separately, with privacy budget $\frac{\Phi}{4}$ and failure probability $\frac{\beta}{4}$, sending the results to the analyzer.

6. The analyzer determines the best estimation of negative/positive part, denoted as $\widetilde{\mathrm{Sum}}(\hat{\mathbf{I}}_s^-)$ and $\widetilde{\mathrm{Sum}}(\hat{\mathbf{I}}_s^+)$. It then combines the information above to give a final sum estimation

$$\widetilde{\mathrm{Sum}}(\mathbf{I}) = (HD)^{-1}\left(\widetilde{\mathrm{Sum}}(\hat{\mathbf{I}}_s^+) - \widetilde{\mathrm{Sum}}(\hat{\mathbf{I}}_s^-) + n \cdot \widetilde{\boldsymbol{m}}\right)$$

In Appendix A.5, we prove the following error bound of the protocol above:

**Theorem 5.3.** *Given $\Phi$, $\beta$, $B$, $d$, assume $n \geq cd\log\frac{\rho_{\max}}{\rho_{\min}}\log\frac{d\log B \log\frac{\rho_{\max}}{\rho_{\min}}}{\beta}k^2\log^4 B$ for some large enough constant c. Then with probability at least $1 - \beta$, the $\ell_2$ error of sum estimation is no greater than*

$$O\left(\sqrt{\log\frac{nd}{\beta}} \min_{s\in\mathbb{R}_{\geq 0}} \left(\sqrt{\sum_u \mathbb{1}(s\sqrt{\Phi(u)/2} < \omega(\mathbf{I}))}\omega(\mathbf{I}) + \sqrt{ndt\ln\frac{td}{\beta}s}\right)\right)$$

*where $t = \lceil\log dB\sqrt{\frac{\rho_{\max}}{\rho_{\min}}}\rceil$.*

## 6 Experiments

In this section, we report the experimental results comparing our new protocols against the baseline method, which adds Gaussian noise of scale $\frac{\tau}{\sqrt{\Phi(u)}}$ after truncating each user's data by a uniform threshold $\tau$. As there is currently no method for choosing a good $\tau$, we give this baseline method the unfair advantage of using the optimal $\tau$ (selected in a non-private manner). We call this baseline the *Naive Optimal*. Note that this baseline is always no worse than the naive method that adds a Gaussian noise of scale $\frac{B}{\sqrt{\Phi(u)}}$ without truncation, which is used in prior work [25, 36, 7, 35, 38], since the latter is the special case of the former with $\tau = B$. The corresponding codes and data are provided in the GitHub repository [2].

### 6.1 Setup

**Datasets.** We performed experiments on both synthetic and real-world datasets. Synthetic datasets are used to demonstrate the scalability of our mechanisms and to examine the effect of different input distributions, varying numbers of users $n$, and different data dimensions $d$. Specifically, for users' data $\mathbf{I}(u)$, we tried two different distributions: In *Normal Data*, each coordinate of each user's data is independently drawn from a normal distribution with mean 1,000 and standard deviation 100. The sampled values are rounded to the nearest integer. In *Uniform Data*, each coordinate of each user's data is uniformly drawn from $\{0, 1, ..., 1000\}$. For both input distributions, we examined various user counts $n = 10^3, 10^4, 10^5, 10^6$, with a default of $10^5$. We also tested different dimensionalities $d \in \{32, 64, 128, 256, 512\}$, with a default value of 128. We set $B = 1,000,000$, which is a sufficiently large upper bound for all datasets.

The real-world data we used is the MNIST (train) dataset [12], which consists of 60,000 images of handwritten digits, where each image is represented by a vector of dimension $28 \times 28 = 784$ and each coordinate is an integer ranging from 0 to 255. We perform sum estimation for each digit separately and treat each image as an individual's data. In order to apply the Hadamard matrix, we

---

[2] https://github.com/personalizedldp/PLCDP

add zeros to the end of each vector to pad the dimension to $d = 1024$ and we set $B = 255\sqrt{d} = 8,160$.

| Data | Result $\ell_2$ Norm | Technique | Relative $\ell_2$ Error(%) | Time(s) |
|---|---|---|---|---|
| Normal Data | $9.04 \times 10^8$ | Naive | 51.24 | 0.02 |
| | | Radius Sum | 9.75 | 0.93 |
| | | Diameter Sum | **0.14** | 16.10 |
| Uniform Data | $5.65 \times 10^8$ | Naive | 52.67 | 0.02 |
| | | Radius Sum | 7.39 | 1.03 |
| | | Diameter Sum | **3.10** | 16.50 |
| MNIST Digit 0 | $4.39 \times 10^7$ | Naive | 85.77 | 0.02 |
| | | Radius Sum | 41.84 | 0.92 |
| | | Diameter Sum | **6.54** | 25.40 |

Table 1: Summary of results under default setting, where $n = 10^5$ and $d = 128$ for synthetic data.

**Privacy Specification.** We used a similar privacy specification as in [2, 20], where we randomly divide the users into two groups: conservative, representing users with high privacy concern; and liberal, representing users with moderate concern. The fraction of users in the conservative and liberal groups are set to $0.05$ and $0.95$, respectively. The privacy level for the users in the conservative and liberal groups are drawn uniformly at random from the ranges $[\frac{1}{n}, 1]$ and $[1, 100]$, respectively, which are reasonable values in the local model of DP according to [8].

**Experimental Parameters.** All experiments are done on a desktop PC equipped with an M2 Pro CPU and 16GB memory. We set the probability parameter $\beta = 0.1$. Each experiment is repeated 20 times and we record the average running time and relative error compared to the true sum. We discard the top/lower $10\%$ errors before computing the average error.

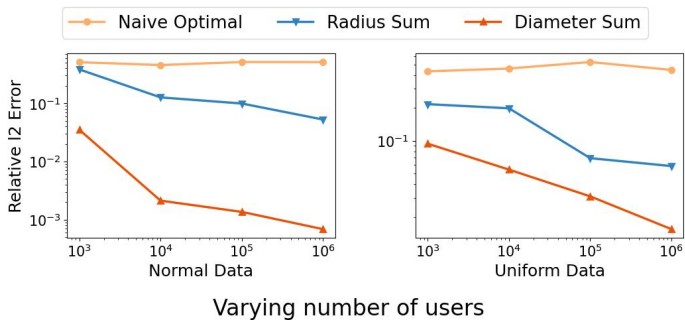

Figure 1: Effect of varying number of users on the relative $\ell_2$ error of different mechanisms.

## 6.2 Results

Table 1 summarizes the results of different mechanisms under the default setting. Our diameter sum mechanism achieves the best performance with acceptable relative error in all cases. In contrast, the naive mechanism always provides poor utility. On the other hand, our diameter sum mechanism always provides improvements compared to the radius sum mechanism, and this improvement becomes essential on the MNIST dataset, whereas the radius sum has a more then $40\%$ error thus loosing utility. Regarding the running time, although there is a large gap between the total running times of different mechanisms, all of them can be efficiently executed on commodity hardware.

**Synthetic Data.** Figure 1 shows the results on the synthetic data varying number of users $n$. As $n$ grows up, the relative error of our radius/diameter sum mechanism decreases roughly linearly in $\sqrt{n}$. This is because for a fixed input distribution and privacy specification, the optimal noise scale $s$ chosen by our algorithms roughly remains unchanged. So the error stated in Theorem 4.1 and Theorem 5.3 roughly grows at the rate of $\sqrt{n}$. Since the $\ell_2$ norm of $\text{Sum}(\mathbf{I})$ is proportional to $n$, thus the relative error will decrease proportional to $\sqrt{n}$. Meanwhile, the relative error of the naive mechanism roughly remains as a constant when $n$ changes. This is because as $n$ grows up, the portion of users with small privacy parameter (thus high error) is fixed. So the total error also grows linearly in $n$ and the relative error remains unchanged.

Figure 2 shows the effect of varying data dimension $d$. We can see as $d$ increases, the error of all mechanisms also increase. The intuition is that the optimal noise scale $s$ is proportional to $\sqrt{d}$. Since the noise vector has $d$ dimensions, the $\ell_2$ norm of each noise vector is proportional to $d$. As the $\ell_2$ norm of $\mathrm{Sum}(\mathbf{I})$ is proportional to $\sqrt{d}$, the relative error roughly increases at the rate of $\sqrt{d}$.

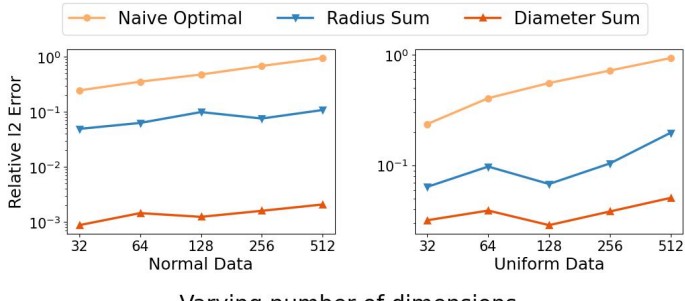

Varying number of dimensions

Figure 2: Effect of varying input dimension $d$ on the relative $\ell_2$ error of different mechanisms.

**MNIST.** Figure 3 shows the accuracy and running time of different mechanisms on the MNIST dataset. Similar to before, the naive mechanism provides the worst performance with almost $100\%$ relative error which makes it meaningless. However, this time our radius sum mechanism also provides poor utility. This is because each digit in the MNIST dataset has its own pattern, thus an error that scales with the radius of the dataset indeed significantly overestimates the sensitivity of each digit and leads to an undesirable error. Fortunately, our diameter sum algorithm is capable of automatically finding the intrinsic pattern of each digit, which is the median vector $\widetilde{\boldsymbol{m}}$ we find in step 3 of the algorithm. Thus it can provide a small error that only scales the diameter of each digit (the variety within the same class). The result on different digits varies slightly, but the diameter sum mechanism provides high accuracy in general. All these mechanisms can be executed efficiently.

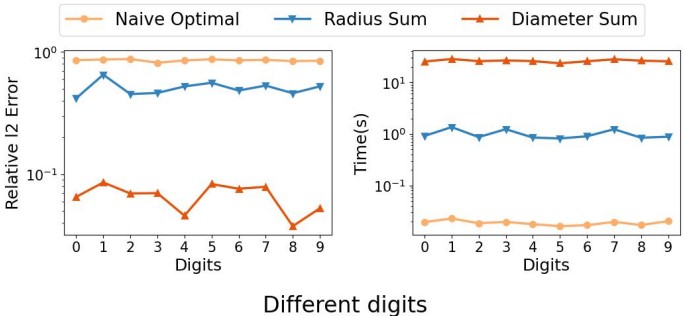

Different digits

Figure 3: MNIST dataset, different digits.

## 7 Limitations and Future Directions

In this paper, we have considered a personalized LDP model where the privacy parameter, i.e., $\Phi(u)$ is known to the analyst. In situations where there is a direct relationship between the user's data and their privacy requirement, such as income data, revealing $\Phi(u)$ would breach privacy. There have been some recent proposals [4, 29] on how to model such a setting where both the data and privacy parameters are to be protected, and it would be interesting to see if our techniques can be extended to this case.

Another interesting direction is to provide a confidence interval (confidence regions in high dimensions), instead of just a sum estimate, which would allow the analyst to make decisions with more statistical reliability. Note that the confidence interval itself must also be differentially private. For the naive Gaussian method, this is easy since the analyst knows precisely the distribution of the noise. However, this is more challenging for our method, and any truncation based approach, because the truncation bias depends on the private data.

## Acknowledgments and Disclosure of Funding

This work is supported by HKRGC under grants 16205422, 16204223, and 16203924. Wei Dong is supported by the NTU–NAP Startup Grant (024584-00001) and the Singapore Ministry of Education Tier 1 Grant (RG19/25). Yuan Qiu is supported by the Start-up Research Fund of Southeast University under grant RF1028625150. Graham Cormode is supported by in part by EPSRC grant EP/V044621/1. We would also like to thank the anonymous reviewers who have made valuable suggestions on improving the presentation of the paper.

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

# A  Appendix A: Proofs

## A.1  Proof of Lemma 3.2

According to Lemma 2.2 in [9], suppose $P$ and $Q$ are distributions on $\Omega \times \Theta$. Let $P'$ and $Q'$ denote the marginal distributions on $\Omega$ induced by $P$ and $Q$ respectively. For $x \in \Omega$, let $P'_x$ and $Q'_x$ denote the conditional distributions on $\Theta$ induced by $P$ and $Q$ respectively, where $x$ specifies the first coordinate. Then

$$\mathrm{D}_\alpha\left(P'\|Q'\right) + \min_{x \in \Omega} \mathrm{D}_\alpha\left(P'_x\|Q'_x\right) \leq \mathrm{D}_\alpha(P\|Q) \leq \mathrm{D}_\alpha\left(P'\|Q'\right) + \max_{x \in \Omega} \mathrm{D}_\alpha\left(P'_x\|Q'_x\right)$$

In our case, let $P$ be the distribution of $\mathcal{M}_2(\mathbf{I}(u), y(u))$ and $Q$ be the distribution of $\mathcal{M}_2(\mathbf{I}'(u), y'(u))$. Here $y'(u)$ denotes the information obtained from $(\mathcal{M}_1(\mathbf{I}(u)))_u$ except for changing user $u$'s output from $\mathcal{M}_1(\mathbf{I}(u))$ to $\mathcal{M}_1(\mathbf{I}'(u))$. Then we have

$$\begin{aligned}
\mathrm{D}_\alpha(P\|Q) &\leq \mathrm{D}_\alpha(\mathcal{M}_2(\mathbf{I}(u)), \mathcal{M}_2(\mathbf{I}'(u))) + \max_{\mathbf{I}(u)} \mathrm{D}_\alpha(y(u), y'(u)) \\
&\leq \Phi_2(u) + \mathrm{D}_\alpha(\mathcal{M}_1(\mathbf{I}(u)), \mathcal{M}_1(\mathbf{I}'(u))) \\
&= \Phi_1(u) + \Phi_2(u)
\end{aligned}$$

Here the second line follows from the post-processing property in Lemma 2.2 of [9], where $y(u)$ can be viewed as a post-processing of $(\mathcal{M}_1(\mathbf{I}(u)))_u$.

## A.2  Proof of Theorem 4.1

Privacy is straightforward. According to Lemma 3.1, each iteration of the for-loop in Algorithm 1 preserves $\frac{\Phi(u)}{t}$-CDP. Since this holds for any $u$, according to the definition of PLCDP, each iteration will be $\frac{\Phi(\cdot)}{t}$-PLCDP. Then the whole process will be $\Phi$-PLCDP according to basic composition.

Next we prove the utility. We first show that with high probability, the $\widetilde{\mathrm{Sum}}_i(\mathbf{I})$ we obtain in each round under-estimates $\mathrm{Sum}(\mathbf{I})$. Conditioned on this, taking $\max$ for $\widetilde{\mathrm{Sum}}(\mathbf{I})$ can only reduce the error.

We should note that the truncated sum is always smaller than the true sum, that is:

$$\mathrm{Sum}(\bar{\mathbf{I}}_i) - \mathrm{Sum}(\mathbf{I}) = \sum_u \min\left(0, \frac{s_i\sqrt{2\Phi(u)}}{\|\mathbf{I}(u)\|_2} - 1\right)\mathbf{I}(u) \preceq \mathbf{0}.$$

The noisy sum $\mathrm{Sum}(\widetilde{\mathbf{I}}_i)$ consists of $n$ i.i.d. Gaussian noises $\mathcal{N}(0, ts_i^2 \cdot \mathbf{1}_{d \times d})$, which can be viewed as a single Gaussian $\mathcal{N}(0, nts_i^2 \cdot \mathbf{1}_{d \times d})$. According to the Gaussian tail bound in Lemma 3.3, with probability at least $1 - \beta$, the magnitude of noise added on any coordinate of $\mathrm{Sum}(\widetilde{\mathbf{I}}_i)$ in any round $i$ is no greater than $s_i\sqrt{2nt\ln\frac{2td}{\beta}}$, which is the amount subtracted by the server in line 4 of Algorithm 2. Conditioned on this, we have for any iteration $i$:

$$\mathbf{0} \preceq \widetilde{\mathrm{Sum}}_i(\mathbf{I}) \preceq \mathrm{Sum}(\bar{\mathbf{I}}_i) \preceq \mathrm{Sum}(\mathbf{I}).$$

Here $\boldsymbol{a} \preceq \boldsymbol{b}$ means $a[j] \leq b[j]$ for each coordinate $j$. That is, each coordinate of any $\widetilde{\mathrm{Sum}}_i(\mathbf{I})$ is an underestimation of the real sum at that coordinate.

Then we show for any choice of $s \in \mathbb{R}_{\geq 0}$, our error is no greater than the value stated in equation (4) for that $s$. First note we only need to consider $s \leq \frac{B}{\sqrt{\rho_{\min}}}$ since a larger $s$ will increase the noise without reducing the truncation error.

When $s = 0$, the value in Equation (4) is $\|\mathrm{Sum}(\mathbf{I})\|_2$, and clearly

$$\|\widetilde{\mathrm{Sum}}(\mathbf{I}) - \mathrm{Sum}(\mathbf{I})\|_2 \leq \|\mathrm{Sum}(\mathbf{I})\|_2.$$

For any other $s$, we can always find an $i$ such that $s_i/2 < s \le s_i$. Then

$$\mathbf{0} \succeq \mathrm{Sum}(\bar{\mathbf{I}}_i) - \mathrm{Sum}(\mathbf{I}) = \sum_u \min\left(0, \frac{s_i\sqrt{2\Phi(u)}}{\|\mathbf{I}(u)\|_2} - 1\right)\mathbf{I}(u)$$

$$\succeq \sum_u \min\left(0, \frac{s\sqrt{2\Phi(u)}}{\|\mathbf{I}(u)\|_2} - 1\right)\mathbf{I}(u)$$

meaning the truncation error for $\mathrm{Sum}(\bar{\mathbf{I}}_i)$ is smaller compared to using $s$ as the truncation threshold. This further implies

$$\|\mathrm{Sum}(\bar{\mathbf{I}}_i) - \mathrm{Sum}(\mathbf{I})\|_2 \le \left\|\sum_u \min(0, \frac{s\sqrt{2\Phi(u)}}{\|\mathbf{I}(u)\|_2} - 1)\mathbf{I}(u)\right\|_2$$

Thus with probability at least $1 - \beta$ we have:

$$\|\widetilde{\mathrm{Sum}}_i(\mathbf{I}) - \mathrm{Sum}(\mathbf{I})\|_2 = \left\|\mathrm{Sum}(\bar{\mathbf{I}}_i) + \mathcal{N}\left(0, nts_i^2 \cdot \mathbf{1}_{d\times d}\right) - s_i\sqrt{2nt\ln\frac{2td}{\beta}} \cdot \mathbf{1} - \mathrm{Sum}(\mathbf{I})\right\|_2$$

$$\le \left\|\mathrm{Sum}(\bar{\mathbf{I}}_i) - \mathrm{Sum}(\mathbf{I})\right\|_2 + 2s_i\sqrt{2ndt\ln\frac{2td}{\beta}}$$

$$\le \left\|\sum_u \min\left(0, \frac{s\sqrt{2\Phi(u)}}{\|\mathbf{I}(u)\|_2} - 1\right)\mathbf{I}(u)\right\|_2 + 4s\sqrt{2ndt\ln\frac{2td}{\beta}}$$

Additionally, since $\widetilde{\mathrm{Sum}}(\mathbf{I})$ is obtained by taking the maximum on each coordinate of under-estimates, we have

$$\|\widetilde{\mathrm{Sum}}(\mathbf{I}) - \mathrm{Sum}(\mathbf{I})\|_2 \le \|\widetilde{\mathrm{Sum}}_i(\mathbf{I}) - \mathrm{Sum}(\mathbf{I})\|_2$$

The above inequality holds for any $s$, so the actual error of Algorithm 2 should be no greater than the minimum of them. The $\ell_\infty$ bound can be obtained similarly, by observing that

$$\|\widetilde{\mathrm{Sum}}_i(\mathbf{I}) - \mathrm{Sum}(\mathbf{I})\|_\infty = \left\|\mathrm{Sum}(\bar{\mathbf{I}}_i) + \mathcal{N}\left(0, nts_i^2 \cdot \mathbf{1}_{d\times d}\right) - s_i\sqrt{2nt\ln\frac{2td}{\beta}} \cdot \mathbf{1} - \mathrm{Sum}(\mathbf{I})\right\|_\infty$$

$$\le \left\|\mathrm{Sum}(\bar{\mathbf{I}}_i) - \mathrm{Sum}(\mathbf{I})\right\|_\infty + 2s_i\sqrt{2nt\ln\frac{2td}{\beta}}$$

$$\le \left\|\sum_u \min\left(0, \frac{s\sqrt{2\Phi(u)}}{\|\mathbf{I}(u)\|_2} - 1\right)\mathbf{I}(u)\right\|_\infty + 4s\sqrt{2nt\ln\frac{2td}{\beta}}$$

### A.3 Answering All Range Queries using Hierarchical Histograms

Here we describe in detail how to adopt the hierarchical histogram approach in [23] together with our PLCDP randomizer/analyzer described in Section 4 to answer all range counting queries. We consider the single-dimension setting, where users' values are integers in $[B]$. Our target is to construct a (private) hierarchical structure that can answer arbitrary range counting queries efficiently and accurately. For clarity, let us assume $B+1$ is a power of 2; otherwise we can just use $\lceil\log(B+1)\rceil$ in place of each $\log(B+1)$ in the following discussion.

Figure 4 provides a graphical illustration of the hierarchical structure when privacy is not involved. In the $h$th level of the hierarchical structure (0 is the highest level and $\log(B+1)$ is the lowest level), the range $[0, B]$ is divided into $2^h$ disjoint bins, each with length $2^{\log(B+1)-h}$. Each user encodes his/her value $\mathbf{I}(u)$ into a frequency vector $\mathbf{H}_h(u) \in \{0,1\}^{2^h}$ indicating which interval his value belongs to. Say,

$$\mathbf{H}_h(u)[j] = \begin{cases} 1, & \text{if } \mathbf{I}(u) \in [j \cdot 2^{\log(B+1)-h}, (j+1) \cdot 2^{\log(B+1)-h} - 1]; \\ 0, & \text{otherwise}. \end{cases}$$

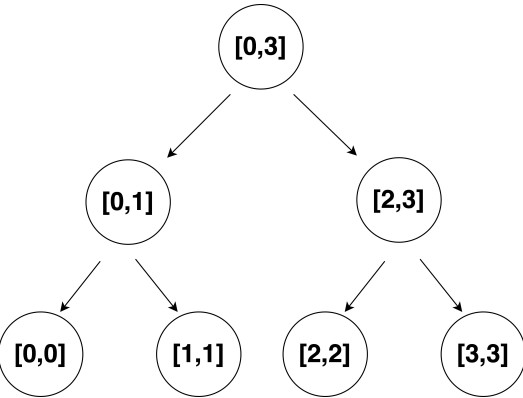

Figure 4: An example with $B = 3$. In the $h$th level, the frequency vector $\mathbf{H}_h(u)$ has dimension $2^h$. Summing up all the $\mathbf{H}_h(u)$s for all users will give a histogram of that level.

It is easy to see a user's data at any level in the hierarchy is one-hot. Under such construction, summing all the $\mathbf{H}_h(u)$'s in each level for all users will give a histogram for that level.

To make the whole process preserve $\Phi$-PLCDP, we invoke Algorithm 1, 2 to obtain a private sum for each level $h = 0, 1, ..., \log(B + 1)$. On the user side, we invoke the LocalSum-r Algorithm with specially designed inputs $\mathbf{H}_h(u)$, $\Phi'$, $B'_h$, $\beta'$, $d'_h$. Here $\mathbf{H}_h(u)$ is constructed as above, with dimension $d'_h = 2^h$ for the $h$th level; $\Phi'(u) = \frac{\Phi(u)}{\log(B+1)}$; $\beta' = \frac{\beta}{\log(B+1)}$ as the privacy/failure probability allocated for each level; and $\beta$ for the failure probability for the whole process. For the value of $B'_h$, since the $\ell_2$ norm of $\mathbf{H}_h(u)$ is exactly 1, we set $B'_h \equiv 1$.

For each level $h$, LocalSum-r$(\mathbf{H}_h(u), \Phi', B'_h, \beta', d'_i)$ returns a set of $t'$ vectors, where $t' = \lceil \log \left( \sqrt{\frac{\rho_{\max}}{\rho_{\min}}} B'_h \right) \rceil = \lceil \log \sqrt{\frac{\rho_{\max}}{\rho_{\min}}} \rceil$ is invariant across different levels. We denote the result returned by LocalSum-r$(\mathbf{H}_h(u), \Phi', B'_h, \beta', d'_h)$ as $\{\widetilde{\mathbf{H}}_{hi}(u)\}_{i=0}^{t'}$, which corresponds to truncated noisy values at different noise scales. And then $\left\{ \{\widetilde{\mathbf{H}}_{hi}(u)\}_{i=0}^{t'} \right\}_{h=0}^{\log(B+1)}$ contains all the required information to build the whole noisy hierarchical histogram, where the inner index $i$ represents different noise scales in that level and the outer index $h$ represents different levels of the hierarchical structure. Details are shown in Algorithm 3.

On the server side, as described in Algorithm 4, we invoke LocalSum-a$(\{\widetilde{\mathbf{H}}_{hi}(u)\}_{i=0}^{t'}, \Phi', B'_h, \beta', d'_h)$ for each level with $\{\widetilde{\mathbf{H}}_{hi}(u)\}_{i=0}^{t'}$ obtained from the user side and parameters $\Phi'$, $B'_h$, $\beta'$, $d'_h$ defined the same as above.

The above procedure will provide us a noisy histogram for each level. Then in order to select the desired quantile, we should use binary search to find the smallest $m$ such that the (noised) frequency of $[0, m]$ is greater than $\frac{n}{2}$. Note any interval $[0, m]$ can be covered with at most $\log(B + 1)$ bins inside the hierarchical structure (to be more specific, at most one bin from each level). Thus the noisy frequency of $[0, m]$ is at most $\log(B + 1)$ times of the maximum error for each bin. Below is the detailed algorithm.

The median selection process can be done in one round with a single message, which is divided into segments where each segment corresponds to a noisy frequency vector. For each level, the user should send $t' = O(\log \frac{\rho_{\max}}{\rho_{\min}})$ message segments and thus $O(\log B \log \frac{\rho_{\max}}{\rho_{\min}})$ message segments in total. The average length of these message segments will be $O(B)$.

To reduce communication complexity, one may apply the sampling technique as in [23], say, each user only randomly picks one level in the hierarchical structure to join. This optimization can reduce the number of message segments by a factor of $\log B$. While this will save privacy it does introduce additional variance, as the final result needs to be scaled back by multiplying $\log B$. The overall effect of sampling is to increase the rank error by a factor of $\sqrt{\log B}$. This is acceptable for our setting when $n$ is large, since what we require is any interior point: we don't really care about the rank error so long as it remains between the minimum and maximum.

---

**Algorithm 3:** LocalHist-r

---

**Input:** $\mathbf{I}(u)$, $\Phi$, $B$, $\beta$

1 $\beta' \leftarrow \frac{\beta}{\log(B+1)}$, $\Phi' \leftarrow \frac{\Phi}{\log(B+1)}$, $t' \leftarrow \lceil \log \sqrt{\frac{\rho_{\max}}{\rho_{\min}}} \rceil$;

2 **for** $h \leftarrow 0, 1, \ldots, \log(B+1)$ **do**

3      $d'_h \leftarrow 2^h$;

4      Define $\mathbf{H}_h(u) \in \{0,1\}^{d'_h}$ such that

$$\mathbf{H}_h(u)[j] \leftarrow \begin{cases} 1, & \text{if } \mathbf{I}(u) \in [j * 2^{\log(B+1)-h}, (j+1) * 2^{\log(B+1)-h} - 1]; \\ 0, & \text{otherwise .} \end{cases}$$

     ;

5      $\{\widetilde{\mathbf{H}}_{hi}(u)\}_{i=0}^{t'} = \mathrm{LocalSum} - \mathrm{r}(\mathbf{H}_h(u), \Phi', 1, \beta', d'_h)$;

6 **end**

7 **return** $\left\{ \{\widetilde{\mathbf{H}}_{hi}(u)\}_{i=0}^{t'} \right\}_{h=0}^{\log(B+1)}$ ;

---

**Algorithm 4:** LocalHist-a

---

**Input:** $\left\{ \{\widetilde{\mathbf{H}}_{hi}(u)\}_{i=0}^{t'} \right\}_{h=0}^{\log(B+1)}$, $\Phi$, $B$, $\beta$

1 $\beta' \leftarrow \frac{\beta}{\log(B+1)}$, $\Phi' \leftarrow \frac{\Phi}{\log(B+1)}$, $t' \leftarrow \lceil \log \sqrt{\frac{\rho_{\max}}{\rho_{\min}}} \rceil$;

2 **for** $h \leftarrow 0, 1, \ldots, \log(B+1)$ **do**

3      $d'_h \leftarrow 2^h$;

4      $\widetilde{\mathrm{Sum}}_h \leftarrow \mathrm{LocalSum} - \mathrm{a}(\{\widetilde{\mathbf{H}}_{hi}(u)\}_{i=0}^{t'}, \Phi', 1, \beta', d'_h)$;

5 **end**

6 **return** $\{\widetilde{\mathrm{Sum}}_h\}_{h=0}^{\log(B+1)}$;

---

In parallel, one may apply the lossless compression technique as in [16], which reduces the size of each message segment from $O(B)$ to $O(\log B + \sqrt{\rho_{\max}})$. The intuition is as follows: since DP definitely induces loss of information, there is no need to send the full information at the beginning. Instead, we can send a seed $s$, which has a much smaller size, and expand it via pseudorandom generators to recover the result. Given an exponentially strong pseudorandom generator and an algorithm that properly chooses $s$, it is demonstrated in [16] that one can reduce the message size significantly while preserving utility. Such reduction requires rejection sampling and thus leads to additional computational costs at the user side. Note this compression technique can be done in parallel with sampling, thus if applying both, the communication cost can be reduced to $O(\log \frac{\rho_{\max}}{\rho_{\min}})$ message segments in total, whereas each segment has size $O(\log B + \sqrt{\rho_{\max}})$.

### A.4 Proof of Theorem 4.3

We start by analyzing the $\ell_\infty$ error of the histograms in each level.

Consider the $h$th level, according to Theorem 4.1, the $\ell_\infty$ error of $\widetilde{\mathrm{Sum}}_h$ is no greater than

$$\min_{s \in \mathbb{R}_{\geq 0}} \left( \| \sum_u \min(0, \frac{s\sqrt{2\Phi'(u)}}{\|\mathbf{H}_h(u)\|_2} - 1)\mathbf{H}_h(u)\|_\infty + 4\sqrt{2nt'\ln\frac{2t'd'_h}{\beta'}s} \right) \quad (7)$$

where $t'$, $\mathbf{H}_h(u)$, $\Phi'$, $B'_h$, $\beta'$, $d'_h$ are defined as in Algorithm 3.

Notably, in the special case of median selection/histogram construction, we always have $\|\mathbf{H}_h(u)\|_2 = 1$, furthermore, the dimension $d'_h$ of the histograms at each level is bounded by $B+1$. Then the guarantee in Equation (7) reduces to

$$\min_{s \in \mathbb{R}_{\geq 0}} \left( \sum_u \mathbb{1}(s\sqrt{2\Phi'(u)} < 1) + 4\sqrt{2nt'\ln\frac{2t'(B+1)}{\beta'}s} \right) \quad (8)$$

With the increase of $s$, the noise increases but the bias reduces when it hits $\frac{1}{\sqrt{2\Phi'(u)}}$. So the minimum of Equation (8) must be obtained at $s = \frac{1}{\sqrt{2\Phi'(u)}}$ for some $u$. Assume values in $\Phi$ are ranked in non-decreasing order, Equation (8) is equivalent to finding

$$\min_{i \in [n]} \left( i + \frac{4}{\sqrt{2\Phi'(u_i)}} \sqrt{2nt'\ln\frac{2t'(B+1)}{\beta'}} \right). \tag{9}$$

Let $k'$ be the minimum $k$ such that $\sum_{i=1}^{k} \sqrt{\Phi'(u_i)} \geq 1$. We show $\min_{i \in [n]} \left( i + \frac{1}{\sqrt{\Phi'(u_i)}} \right) \leq 4k'$ in Lemma A.1. As a result, the $\ell_\infty$ error of the histogram at each level is bounded by $O(\sqrt{nt'\ln\frac{t'B}{\beta'}}k')$ with probability $1 - \beta$. By taking a union bound, this leads to an error of

$$O\left( \sqrt{nt'\ln\frac{t'B\log B}{\beta'}}k' \right)$$

which holds simultaneously for histograms in all levels.

To move from the error of histograms to the error of range queries, we should note that each range query can be obtained by summing at most $2\log B$ entries of the histograms. Thus the error of each query is bounded by

$$O\left( \sqrt{nt'\ln\frac{t'B\log B}{\beta'}}k'\log B \right)$$

Finally, plug in the values of $t'$ and $\beta'$ chosen in the previous subsection. For $k'$, note that $k' \leq \sqrt{\log B}k$, where $k$ is the smallest index such that $\sum_{i=1}^{k} \sqrt{\Phi(u_i)} \geq 1$. The error is therefore bounded by

$$O\left( \sqrt{n\log\frac{\rho_{\max}}{\rho_{\min}}\log\frac{\log B\log\frac{\rho_{\max}}{\rho_{\min}}}{\beta}}k\log^2 B \right)$$

Below we complete the proof of Lemma A.1

**Lemma A.1.** *Assume values in $\Phi$ are ranked in non-decreasing order and $\rho_{\min} > \frac{1}{n^2}$. Let $k$ be the smallest index such that $\sum_{i=1}^{k} \sqrt{\Phi(u_i)} \geq 1$, then*

$$\min_{i \in [n]} \left( i + \frac{1}{\sqrt{\Phi(u_i)}} \right) \leq 8k$$

*Proof.* Let $i^* = \operatorname{argmin}_i\{i + \frac{1}{\sqrt{\Phi(u_i)}}\}$, and let $M = \max\{i^*, \frac{1}{\sqrt{\Phi(u_i^*)}}\}$.

Consider $i' = \lfloor \frac{M}{2} \rfloor$. We have by definition

$$i' + \frac{1}{\sqrt{\Phi(u_i')}} \geq i^* + \frac{1}{\sqrt{\Phi(u_i^*)}} > M \geq 2i'$$

So $i'\sqrt{\Phi(u_i')} < 1$. Since $\Phi$ is ordered, all $\Phi(u_i)$ with $i \leq i'$ are no greater than $\Phi(u_i')$. As a result, we have

$$\sum_{i=1}^{i'} \sqrt{\Phi(u_i)} \leq i'\sqrt{\Phi(u_i')} < 1$$

which means $k \geq i'$. On the other hand, we have

$$\min_{i \in [n]} \left( i + \frac{1}{\sqrt{\Phi(u_i)}} \right) \leq 2M \leq 8i' \leq 8k$$

which completes the proof. $\qquad\square$

## A.5 Omitted details in Section 5

Before going to the proofs, we first briefly introduce the intuition behind the proof. First of all, with high probability Step 1 can provide a 'good rotation' such that for any pair of users' data, Lemma 5.2 holds. Then with high probability Step 2&3 can find an interior point of the rotated dataset on each dimension. Conditioned on these two events, with high probability the radius of the shifted data $\hat{\mathbf{I}}_s$ is $\tilde{O}(\sqrt{d}\omega(\mathbf{I}))$. Note that when rotating back by multiplying $(HD)^{-1}$ in the last step, the $\ell_2$ norm of the result will be decreased by a factor of $\frac{1}{\sqrt{d}}$ so the additional $\sqrt{d}$ factor here will be eliminated finally. That is, we can perform the sum algorithm safely.

Since the shifted data involves randomness depending on both random rotation and median selection, to obtain a deterministic error bound as stated in Theorem 5.3, we aim to characterize the worst case. This happens when the shifted data is obtained by subtracting the minimum (maximum) value of each dimension. We denote such datasets as $\hat{\mathbf{I}}_s^{+*}(\hat{\mathbf{I}}_s^{-*})$ respectively. These datasets are the 'worst' in the sense that their positive/negative part has the largest radius and leads to the largest error, which is proved in Lemma A.2. We further analyze the error on these worst case instances in Lemma A.3. Finally, combining all the above arguments and adding up the error of positive/negative parts leads to a complete proof of the main theorem. Below we first present the two lemmas.

For clarity of expression, we denote $\mathrm{Err}(\mathbf{I})$ to be the error of Algorithm 1, 2 when invoked on dataset $\mathbf{I}$ with privacy parameter $\frac{\Phi}{4}$, failure probability $\frac{\beta}{4}$, domain bound $B\sqrt{d}$ and dimension $d$. To be specific, define

$$\mathrm{Err}(\mathbf{I}) = \min_{s \in \mathbb{R}_{\geq 0}} \left( \left\| \sum_u \left( \|\mathbf{I}(u)\|_2 - s\sqrt{\Phi(u)/2} \right)^+ \frac{\mathbf{I}(u)}{\|\mathbf{I}(u)\|_2} \right\|_2 + 4\sqrt{2ndt\ln\frac{8td}{\beta}s} \right) \quad (10)$$

where $t = \lceil \log dB\sqrt{\frac{\rho_{\max}}{\rho_{\min}}} \rceil$.

**Lemma A.2.** *Given $\hat{\mathbf{I}}$, conditioned on the fact that Steps 2–3 correctly find an interior point on each dimension, we have*

$$\mathrm{Err}(\hat{\mathbf{I}}_s^+) \leq \mathrm{Err}(\hat{\mathbf{I}}_s^{+*})$$

*where $\hat{\mathbf{I}}_s^{+*}$ is obtained by subtracting the minimum value of each dimension so that all values inside are non-negative. The same property also holds for $\mathrm{Err}(\hat{\mathbf{I}}_s^-)$.*

*Proof.* First of all, it is easy to see $\hat{\mathbf{I}}_s^+(u) \leq \hat{\mathbf{I}}_s^{+*}(u)$ (on each coordinate) for any $u$. This is because, for each coordinate of $\hat{\mathbf{I}}_s^{+*}(u)$, its corresponding value in $\hat{\mathbf{I}}_s^+(u)$ is either $0$ (moved to negative part) or smaller (since $\hat{\mathbf{I}}_s^{+*}$ subtracts the minimum on each dimension). And this further implies $\|\hat{\mathbf{I}}_s^+(u)\|_2 \leq \|\hat{\mathbf{I}}_s^{+*}(u)\|_2$.

To show $\mathrm{Err}(\hat{\mathbf{I}}_s^+) \leq \mathrm{Err}(\hat{\mathbf{I}}_s^{+*})$, we only need to prove the former has a smaller bias for any $s$ (scale), that is

$$\left\| \sum_u \left( \|\hat{\mathbf{I}}_s^+(u)\|_2 - s\sqrt{\Phi(u)/2} \right)^+ \frac{\hat{\mathbf{I}}_s^+(u)}{\|\hat{\mathbf{I}}_s^+(u)\|_2} \right\|_2 \leq \left\| \sum_u \left( \|\hat{\mathbf{I}}_s^{+*}(u)\|_2 - s\sqrt{\Phi(u)/2} \right)^+ \frac{\hat{\mathbf{I}}_s^{+*}(u)}{\|\hat{\mathbf{I}}_s^{+*}(u)\|_2} \right\|_2$$

Indeed we can show a stronger statement:

$$\sum_u \max\left( 0, 1 - \frac{s\sqrt{\Phi(u)/2}}{\|\hat{\mathbf{I}}_s^+(u)\|_2} \right) \hat{\mathbf{I}}_s^+(u) \leq \sum_u \max\left( 0, 1 - \frac{s\sqrt{\Phi(u)/2}}{\|\hat{\mathbf{I}}_s^{+*}(u)\|_2} \right) \hat{\mathbf{I}}_s^{+*}(u)$$

Because for any $u$, $\max\left( 0, 1 - \frac{s\sqrt{\Phi(u)/2}}{\|\hat{\mathbf{I}}_s^+(u)\|_2} \right) \leq \max\left( 0, 1 - \frac{s\sqrt{\Phi(u)/2}}{\|\hat{\mathbf{I}}_s^{+*}(u)\|_2} \right)$ and all items are non-negative.

For the $\hat{\mathbf{I}}_s^-$ counterpart, one may construct $\hat{\mathbf{I}}_s^{-*}$ by subtracting the maximum value of each dimension and then taking absolute so that all values inside are non-negative. $\square$

**Lemma A.3.** *Conditioned on the fact that the rotated data $\hat{\mathbf{I}}$ satisfies Lemma 5.2 for all pairs of points,*

$$\mathrm{Err}(\hat{\mathbf{I}}_s^{+*}) = O\left(\sqrt{d\log\frac{nd}{\beta}}\min_{s\in\mathbb{R}_{\geq 0}}\left(\sqrt{\sum_u \mathbb{1}(s\sqrt{\Phi(u)/2} < \omega(\mathbf{I}))}\omega(\mathbf{I}) + \sqrt{ndt\ln\frac{td}{\beta}s}\right)\right)$$

*with probability at least $1 - \frac{\beta}{4}$ where $t = \lceil\log dB\sqrt{\frac{\rho_{\max}}{\rho_{\min}}}\rceil$. The same bound also holds for $\mathrm{Err}(\hat{\mathbf{I}}_s^{-*})$.*

*Proof.* We only prove the statement for $\mathrm{Err}(\hat{\mathbf{I}}_s^{+*})$ as the other part follows similarly. According to the definition in Equation (10), we have

$$\mathrm{Err}(\hat{\mathbf{I}}_s^{+*}) = O\left(\min_{s\in\mathbb{R}_{\geq 0}}\left(\left\|\sum_u\left(\|\hat{\mathbf{I}}_s^{+*}(u)\|_2 - s\sqrt{\Phi(u)/2}\right)^+ \frac{\hat{\mathbf{I}}_s^{+*}(u)}{\|\hat{\mathbf{I}}_s^{+*}(u)\|_2}\right\|_2 + \sqrt{ndt\ln\frac{td}{\beta}s}\right)\right) \tag{11}$$

Considering the truncation error term $\left\|\sum_u\left(\|\hat{\mathbf{I}}_s^{+*}(u)\|_2 - s\sqrt{\Phi(u)/2}\right)^+ \frac{\hat{\mathbf{I}}_s^{+*}(u)}{\|\hat{\mathbf{I}}_s^{+*}(u)\|_2}\right\|_2$, we have

$$\left\|\sum_u\left(\|\hat{\mathbf{I}}_s^{+*}(u)\|_2 - s\sqrt{\Phi(u)/2}\right)^+ \frac{\hat{\mathbf{I}}_s^{+*}(u)}{\|\hat{\mathbf{I}}_s^{+*}(u)\|_2}\right\|_2$$

$$\leq\sqrt{\sum_u \mathbb{1}(s\sqrt{\Phi(u)/2} < \|\hat{\mathbf{I}}_s^{+*}(u)\|_2)\mathrm{rad}(\hat{\mathbf{I}}_s^{+*})}$$

$$\leq\sqrt{\sum_u \mathbb{1}(s\sqrt{\Phi(u)/2} < \omega(\mathbf{I})\sqrt{d\log\frac{nd}{\beta}})\omega(\mathbf{I})\sqrt{d\log\frac{nd}{\beta}}}$$

Here the second line is because when the rotation is 'good', the radius of $\hat{\mathbf{I}}_s^{+*}$ is no greater than $\omega(\mathbf{I})\sqrt{d\log\frac{nd}{\beta}}$. Plugging it back to Equation (11) and substituting $s = s' * \sqrt{d\log\frac{nd}{\beta}}$ gives the desired result. Note this error expression is only related to the original data $\mathbf{I}$ and does not involve the randomness on rotation and shift. $\qquad\square$

Below is the complete proof of Theorem 5.3.

*Proof.* Privacy follows from the composition theorem, so we focus on utility here.

With probability at least $1 - \frac{\beta}{4}$, the rotation satisfies Lemma 5.2 for all pairs of points. Further, according to Theorem 4.3, with (another) probability at least $1 - \frac{\beta}{4}$, we can find an interior point in each dimension. This means that Lemma A.2, A.3 hold together with probability at least $1 - \frac{\beta}{2}$.

$$\|\mathrm{Sum}(\mathbf{I}) - \widetilde{\mathrm{Sum}}(\mathbf{I})\|_2$$

$$=\|\mathrm{Sum}(\mathbf{I}) - (HD)^{-1}\left(\widetilde{\mathrm{Sum}}(\hat{\mathbf{I}}_s^+) - \widetilde{\mathrm{Sum}}(\hat{\mathbf{I}}_s^-) + n\cdot\widetilde{\boldsymbol{m}}\right)\|_2$$

$$=\|(HD)^{-1}\left(\widetilde{\mathrm{Sum}}(\hat{\mathbf{I}}_s^+) - \mathrm{Sum}(\hat{\mathbf{I}}_s^+) - \widetilde{\mathrm{Sum}}(\hat{\mathbf{I}}_s^-) + \mathrm{Sum}(\hat{\mathbf{I}}_s^-)\right)\|_2$$

$$\leq\frac{1}{\sqrt{d}}\left(\mathrm{Err}(\hat{\mathbf{I}}_s^+) + \mathrm{Err}(\hat{\mathbf{I}}_s^-)\right)$$

$$\leq\frac{1}{\sqrt{d}}\left(\mathrm{Err}(\hat{\mathbf{I}}_s^{+*})) + \mathrm{Err}(\hat{\mathbf{I}}_s^{-*})\right)$$

$$=O\left(\sqrt{\log\frac{nd}{\beta}}\min_{s\in\mathbb{R}_{\geq 0}}\left(\sqrt{\sum_u \mathbb{1}(s\sqrt{\Phi(u)/2} < \omega(\mathbf{I}))}\omega(\mathbf{I}) + \sqrt{ndt\ln\frac{td}{\beta}s}\right)\right),$$

where the third line is because

$$
\begin{aligned}
\mathrm{Sum}(\mathbf{I}) &= (HD)^{-1}\,\mathrm{Sum}(\hat{\mathbf{I}}) \\
&= (HD)^{-1}\left(\mathrm{Sum}(\hat{\mathbf{I}}_s) + n\cdot\widetilde{\boldsymbol{m}}\right) \\
&= (HD)^{-1}\left(\widetilde{\mathrm{Sum}}(\hat{\mathbf{I}}_s^+) - \widetilde{\mathrm{Sum}}(\hat{\mathbf{I}}_s^-) + n\cdot\widetilde{\boldsymbol{m}}\right)
\end{aligned}
$$

The forth line is because multiplying by $(HD)^{-1}$ decreases the $\ell_2$ norm by a factor of $\frac{1}{\sqrt{d}}$ And the last line comes from applying Lemma A.3 twice on both $\hat{\mathbf{I}}_s^{+*}$ and $\hat{\mathbf{I}}_s^{-*}$, each consumes failure probability of $\frac{\beta}{4}$. Combining all arguments together with probability at least $1-\beta$ the error bound holds.

$\square$

Dividing the above error by $n$ leads to the mean estimation error:

**Corollary A.4.** *Given* $\Phi$, $\beta$, $B$, $d$, $\mathbf{I}$*, if* $n \geq cd\log\frac{\rho_{\max}}{\rho_{\min}}\log\frac{d\log B\log\frac{\rho_{\max}}{\rho_{\min}}}{\beta}k^2\log^4 B$*, there is a* $\Phi$-*PLCDP Algorithm that estimates the mean of* $\mathbf{I}$ *with* $\ell_2$ *error at most*

$$
O\left(\frac{\sqrt{\log\frac{nd}{\beta}}}{n}\min_{s\in\mathbb{R}_{\geq 0}}\left(\sqrt{\sum_u \mathbb{1}(s\sqrt{\Phi(u)/2} < \omega(\mathbf{I}))\omega(\mathbf{I})} + \sqrt{ndt\ln\frac{td}{\beta}s}\right)\right)
$$

*with probability at least* $1-\beta$*, where* $t = \lceil\log dB\sqrt{\frac{\rho_{\max}}{\rho_{\min}}}\rceil$*.*

The whole process is a two-round protocol, where the first round finds an interior point and does the shift, and the second round computes the sum estimation. As discussed in the previous section, without loss of utility, the communication cost of the first round can be reduced to $O(d\log\frac{B\rho_{\max}}{\rho_{\min}})$ message segments per user, whereas each message has size $O(\log B + \sqrt{\rho_{\max}})$. And for the second round, each user sends $t = O(\log\frac{Bd\rho_{\max}}{\rho_{\min}})$ message segments, each with length $O(d)$ (or $O(\log d + \sqrt{\rho_{\max}})$ if $d$ is too large).

