# OpenReview forum: "Sum Estimation under Personalized Local Differential Privacy"
_NeurIPS.cc/2025/Conference — NeurIPS 2025 poster_

### Official Review · Reviewer_bB2d · 2025-06-23

**Clarity:** 3
**Significance:** 2
**Originality:** 3
**Rating:** 4
**Confidence:** 4

**Summary:**

Private sum estimation protocols generally aim to balance a bias term caused by clipping with a term caused by the noise ensuring differential privacy. The best choice of the clipping parameter (and, in some cases, the center of the clipping ball) is highly data dependent. Methods such as Coinpress have been devised to address this tradeoff in the central model of DP.

This paper addresses local differential privacy (LDP) with individual privacy budgets (personalized DP). It suggests a protocol that refines the baseline of choosing a global clipping threshold used for by all local randomizers. The idea is to release each number with multiple clipping thresholds (splitting the privacy budget evenly), spaced geometrically such that a logarithmic number of thresholds covers all relevant thresholds within a constant factor.

There is also a 2-round algorithm that is able to get a bound in terms of the diameter of the dataset, mirroring past work on adaptive mean estimation.

**Questions:**

- Instead of releasing a report for every chosen threshold, did you consider sampling a random one? This might give a better privacy-utility trade-off, in particular for pure DP
- Did you consider what your techniques can do in the context of shuffling or secure aggregation?
- Is there a reason you did not include the inverse sensitivity-based exponential mechanism of [20] as a baseline?

**Ethical Concerns:**

["NO or VERY MINOR ethics concerns only"]

**Final Justification:**

In view of the discussion, in particular the fact that there are no attractive alternatives in the central model, I am now more positively inclined.

**Limitations:**

Most limitations are adequately discussed, but the limitations of LDP algorithms for ML applications could have deserved some discussion.

**Quality:**

3

**Strengths And Weaknesses:**

Sum (and mean) estimation is a key part of many machine learning algorithms, with private variants used in stochastic gradient descent. The proposed algorithm is rather nice and offers a nontrivial guarantee.

On the other hand, for many ML applications the amount of noise of LDP protocols is simply too large, and this is inherent to LDP. Therefore cryptographic techniques such as shuffling and secure aggregation are used to get closer to central model utility guarantees without collecting data. As such, the proposed protocol might have limited impact in machine learning.

---

> ### Author Rebuttal · Authors · 2025-07-27
>
> Reply to Question \#1:
>
> In our protocol, we consider $t= \log \left( \sqrt{\frac{\rho_{\max} }{\rho_{\min}}} B \right) $ thresholds and wants to find the best among them, which requires releasing all the $t$ results and thus we have to divide privacy into $t$ parts.
>
> If we opt not to report each threshold and instead randomly select one, we can save privacy by a factor of $t$. However, the risk of choosing a suboptimal threshold can result in significant overhead, ultimately becoming the dominating factor.
>
> Consider the data in Example 1.1, our radius protocol achieves an error of $O(n^{4\over 3}\log (nB))$ with constant probability. However, if we randomly choose a threshold, the error is likely to exceed $(nB)^{2\over 3}$ with constant probability, which is considerably larger in general.
>
>
> Reply to Question \#2 and Weakness \#1:
>
> We agree that the LDP model introduces larger noise compared to other DP models.  However, because of its simplicity, both the research community and the industry [B] have invested a lot of effort in studying LDP, and this work is therefore important to NeurIPS to fully understand what can (and what cannot) be achieved under this model. In particular, the sum estimation problem under LDP has been extensively studied as a foundational problem [C,D,references in the paper].
>
> You are right that secure computation can often reduce the noise, by using cryptographic techniques to hide some intermediate results against a computationally bounded adversary, at the expense of higher computation/communication cost and implementation complexity. On the other hand, an LDP protocol reveals all intermediate results to a computationally unbounded adversary.  This two approaches are in some sense orthogonal and can be often combined [E], yielding a trade-off between cost and noise. This is a promising direction to combine our LDP protocol with secure computation.
>
> However, our technique cannot be applied to the shuffle model. The key reason is that differing privacy parameters among users negate the`hiding among the crowd' effect, which is essential for the utility amplification of the shuffle model.
>
>
> [B] Cormode, Graham, et al. "Privacy at scale: Local differential privacy in practice." Proceedings of the 2018 international conference on management of data. 2018.
>
> [C] Liu, Junxu, et al. "Cross-silo federated learning with record-level personalized differential privacy." Proceedings of the 2024 on ACM SIGSAC Conference on Computer and Communications Security. 2024.
>
> [D] Duchi, John, and Ryan Rogers. "Lower bounds for locally private estimation via communication complexity." Conference on Learning Theory. PMLR, 2019.
>
> [E] Kairouz, Peter, Ziyu Liu, and Thomas Steinke. "The distributed discrete gaussian mechanism for federated learning with secure aggregation." International Conference on Machine Learning. PMLR, 2021.
>
>
> Reply to Question \#3:
>
> The exponential mechanism (EM) requires computing a utility score (which depends on users' private data) for each possible output. In the central model, this score can be directly computed by the central server.
>
> However, in the local DP model, calculating utility scores necessitates interactions between users. Consequently, the computed utility scores must include DP noise.
> This makes EM meaningless in the LDP model since its utility guarantee is highly sensitive to the accuracy of the utility scores.
> To the best of our knowledge, no prior work has successfully applied the exponential mechanism in the local DP model.

---

> > ### Comment · Reviewer_bB2d · 2025-08-04
> > **Follow-up on rebuttal**
> >
> > Thanks for the answers.
> >
> > Regarding question 1 the suggestion was not to sample a single random threshold, but rather that each LDP report would independently sample a random threshold. Aggregating all reports would then give information on every threshold. I believe this could improve time, utility, or both in the case of pure DP.
> >
> > Regarding question 3 my thinking was that it would be interesting to compare to a central model method.

---

> > > ### Author Response · Authors · 2025-08-05
> > >
> > > Question 1: Thanks for the clarification.  Now we understand your point.  You are right that our algorithm can be further improved using the sampling technique, which has also been used in the standard LDP model [F,G,H]. Under pure DP, this technique can reduce the privacy consumption by a factor of $O(\log B)$, which translates into an improvement of $O(\sqrt{\log B})$ in the error (after accounting for the sampling error).  But under zCDP, which is setting of our paper as we focus on high dimensions, there will be no asymptotic improvement because the error scales with $\sqrt{\rho}$. Nevertheless, this technique can still reduce the message size of each user by an $O(\log B)$ factor.  We will definitely add this extension to the paper. Thanks a lot for the insight.
> > >
> > > Question 3: We agree that comparing our methods with the central model of PDP would offer valuable insights on the level of degradation in utility in exchange for the stronger privacy guarantee offered by the local model.  However, there are no practical high-dimensional sum estimation protocols under the central model of PDP.  The inverse sensitivity-based mechanism you mentioned has been adapted to the central PDP model in [20], but it takes exponential time (even in the 1D case) for the sum estimation problem.  This is because computing the utility scores under PDP in general requires enumerating all possible subsets of the original dataset.  It is an interesting problem to see if one can make the inverse sensitivity mechanism run in polynomial time under the central model of PDP for concrete problems like the sum/mean estimation.
> > >
> > >
> > > [F] Raef Bassily, Kobbi Nissim, Uri Stemmer, Abhradeep Guha Thakurta: Practical Locally Private Heavy Hitters. NIPS 2017: 2288-2296
> > >
> > > [G] Tianhao Wang, Bolin Ding, Jingren Zhou, Cheng Hong, Zhicong Huang, Ninghui Li, Somesh Jha: Answering Multi-Dimensional Analytical Queries under Local Differential Privacy. SIGMOD Conference 2019: 159-176
> > >
> > > [H] Tianhao Wang, Ninghui Li, Somesh Jha: Locally Differentially Private Heavy Hitter Identification. IEEE Trans. Dependable Secur. Comput 2021: 982-993

---

> > > > ### Comment · Reviewer_bB2d · 2025-08-05
> > > >
> > > > Thanks for considering the sampling technique, which is indeed a known technique, and acknowledging that it can improve the algorithm.
> > > >
> > > > About pure DP versus zCDP, could one not invoke the transformation of Bun et al. (https://arxiv.org/abs/1711.04740) to get the same error under pure DP as under zCDP?
> > > >
> > > > Good point about the lack of computational efficiency in the central model, this indeed makes progress in other models more interesting.
> > > >
> > > > In view of this, I will increase my score.

---

> > > > > ### Author Response · Authors · 2025-08-05
> > > > >
> > > > > Thanks for pointing to this result.  Yes, we believe it can be applied in our setting in principle.  However, the precise quantitative result is less clear under our personalized setting.  In particular, Theorem 6.1 in that paper depends on a single $\varepsilon$; in the PDP setting, it should depend on all the $\varepsilon_i$.  How to generalize this result to the PDP setting is an interesting problem to look into.

---

### Official Review · Reviewer_Ku3C · 2025-07-02

**Clarity:** 3
**Significance:** 3
**Originality:** 2
**Rating:** 4
**Confidence:** 3

**Summary:**

This paper studies the problem of sum/mean estimation under personalized LDP, where the users have different privacy parameters (given by a function $\Phi(u)$ for user $u$). They consider specifically the zero-concentrated variant of personalized LDP and assume the elements are in $[B]$. Their methods are based on truncating the elements and adding noise with scale proportional to the truncation threshold to each element. The truncation threshold is of the form $s\sqrt{\Phi(u)}$ for each user $u$, where the universal factor $s$ is determined by the "subtract-max" technique [13] which involves running $\log(B\sqrt{\rho_{\text{max}}/ \rho_{\text{min}}})$  instances of the truncation mechanism each with a different $s$ ($s$ ranging from from $1/\sqrt{2\rho_{\text{max}}})$ to $B/\sqrt{2\rho_{min}})$, and finding the noisy sum that approximately achieves the best balance among the bias and noise. They also give a method with diameter-dependent error, which applies the shift-and-rotate technique from [18] together with the truncation protocol mentioned above.

**Questions:**

- Maybe the authors can highlight any contributions/insights in the paper that are not obvious but are of importance/interest. E.g. why the truncation threshold function is of this particular form, and what insights it brings, etc., or anything else they want to highlight.

**Ethical Concerns:**

["NO or VERY MINOR ethics concerns only"]

**Final Justification:**

My impression of the paper remains largely the same.

**Limitations:**

Yes.

**Paper Formatting Concerns:**

This submission appears to be a few lines over the nine-page limit.

**Quality:**

3

**Strengths And Weaknesses:**

strengths:
- This paper studies an important problem.
- The presented methods are natural.

weaknesses/comments
- The paper might not be easy to read for those not familiar with the related work or DP mean estimation. However, for those who are familiar with these works, they would probably notice that the method presented in this paper uses a combination of existing techniques. In particular, the presented results - one being radius-dependent, the other being diameter-dependent - are consequences of extending the methods in [18] to the personalized privacy setting.
Thus, there is not much novelty in terms of the techniques presented in this paper, which is not a big issue per se, but as it stands there does not appear to be other noteworthy insights/interesting points.
- I do not have a problem with the results in the paper in general, but this paper appears borderline to me.
- Also, this submission appears to be a few lines over the 9-page limit which violates the NeurIPS formatting requirements. Taken this into account, I am leaning more toward the negative end of the border.

---

> ### Author Rebuttal · Authors · 2025-07-27
>
> Reply to Weakness \#1:
>
> The shift-and-truncate framework in [18] is a standard approach for addressing the sum estimation problem, which is not the focus or contribution of our work.
>
> In this paper, we consider the personalized local DP setting, where the error depends on both users' **data** and users' **privacy**. The main challenge lies in determining **how** to perform truncation in a way that balances the effects of these two factors.
>
> To address this, we propose a novel searching strategy that optimizes over both dimensions simultaneously by considering the noise scale $s$. In contrast, [18] relies on a pre-computed quantile for truncation and does not involve any optimization procedures.
>
> Furthermore, the method in [18] cannot be extended to our setting because the target quantile (and thus the truncation threshold) cannot be computed while adhering to the personalized DP requirements. A comparable baseline is the *naive optimal protocol* used in our experiments, which assumes we already know (in a non-private manner) the optimal truncation threshold. Even with this unrealistic advantage, the naive optimal protocol is outperformed by our approach, highlighting the effectiveness and significance of our new strategy.
>
> Reply to Weakness \#3:
>
> We sincerely apologize for our oversight. A minor change was made to the paper shortly before the submission deadline, and we did not realize it caused the paper to exceed the 9-page limit. We'll definitely fix this problem and hope this will not be the reason for rejection. Thanks!
>
> Reply to Question \#1:
>
> Thanks for the comment. As we have explained in our response to Weakness \#1, the main idea and contribution of this paper lie in the proposed searching strategy that considers $\mathbf{I}(u)$ and $\Phi(u)$ simultaneously. That's why the truncation threshold is of the form $\tau_i(u) =s_i \sqrt{2\Phi(u)} = 2^{i}\sqrt{\frac{ \Phi(u) }{\rho_{\max}}}$.
> We will emphasize the core contributions of our work and provide additional explanations to clarify the insights behind them.

---

> > ### Comment · Reviewer_Ku3C · 2025-08-05
> >
> > Given that the setting in this paper is for personalized DP, it makes sense that the truncation threshold would take a different form. Thank you for your response.

---

### Official Review · Reviewer_EmUR · 2025-07-03

**Clarity:** 3
**Significance:** 3
**Originality:** 3
**Rating:** 5
**Confidence:** 2

**Summary:**

This paper provides two protocols for sum/mean estimation in a personalized local differential privacy model, specifically where each user is allowed to set their own privacy parameter to any positive real number. The first protocol has error which scales with the radius or maximum $l_2$ norm of the user's dataset, and the second protocol scales with the diameter of the data. Their results are numerically supported by comparison with a baseline method of adding a user dependent Gaussian noise after the truncating each user's data by a uniform threshold.

**Questions:**

1. It would be interesting to see a theoretical comparison of the naive optimal protocol (section 6) with the radius/diameter dependent protocols, and for which distributions the latter ones are expected to significantly outperform the naive protocol. Are these distributions expected to be similar to various real world/synthetic datasets?
2. Do the protocols described achieve optimal error, if known? It would be helpful to have a discussion regarding the optimality of the described protocols within the first two sections.
3. The formatting of the description of the radius and diameter dependent protocols are inconsistent. It would improve clarity to follow a similar format for both the sections, ideally using the algorithm environment since it is more standard.

**Ethical Concerns:**

["NO or VERY MINOR ethics concerns only"]

**Final Justification:**

My questions were adequately addressed, so I will be keeping my current score

**Limitations:**

yes

**Quality:**

4

**Strengths And Weaknesses:**

The paper is clearly written, and provides detailed proofs with corresponding analysis. The main contribution of this work is to introduce and analyze an interesting model of local differential privacy. The radius dependent protocol under the assumption of a uniform privacy parameter for all users reduces to the optimal mean estimation bounds in prior work by (Huang et. al., 2021. "Instance-optimal mean estimation under differential privacy"). The diameter dependent protocol naturally generalizes this protocol by applying the previous radius dependent guarantee to a 'shifted' origin such that the radius of the shifted dataset is approximately the diameter of the original one.

The numerical results compare the new protocols to a naive optimal protocol, which appears in previous work. However, this choice of baseline could be better justified. It would also be helpful to see a more formal analysis of the cases where the diameter dependent protocol is expected to perform better than the radius dependent protocol.

---

> ### Author Rebuttal · Authors · 2025-07-27
>
> Reply to Question \#1:
>
> We agree that doing a theoretical comparison under some common distributions can complement our experimental observations.
>
>
> We have done some analyses on 1D data for the following two distributions (please let us know if you are interested in other distributions):
>
> (1) Uniform data in the range $[0,n]$.
> (2) Gaussian data with mean $n$ and variance $O(n)$.
>
> Under the same privacy specification as in our experiments (say, 5\% privacy parameters are uniform in $[\frac{1}{n}, 1]$ while the remaining are uniform in $[1, 100]$).
> We can show that, for both data distributions, the error of the naive optimal protocol is $O(n^2)$ while our radius-dependent protocol has error $\widetilde{O}(n^{5/3})$.  For Gaussian data, our diameter-dependent protocol further reduces the error to $\widetilde{O}(n^{3\over 2})$ because the data is clustered away from the origin.
>
>
> From these observations, it is evident that for common real-world data distributions, our methods outperform the naive optimal protocol. Furthermore, the performance gap becomes even more pronounced as the data becomes more concentrated.
>
> Finally, we would like to emphasize that the naive optimal protocol assumes that the optimal truncation threshold is given, which is not easy to obtain under privacy constraints.
>
>
> Reply to Question \#2:
>
> Please see our response to Reviewer \#1 Q1.
>
> Reply to Question \#3:
>
> We agree.  We'll rewrite our diameter-dependent protocol using the algorithm environment to improve clarity.

---

> > ### Comment · Reviewer_EmUR · 2025-08-05
> >
> > Thank you for your response, and for adequately addressing my questions

---

### Official Review · Reviewer_fj6z · 2025-07-22

**Clarity:** 3
**Significance:** 2
**Originality:** 2
**Rating:** 4
**Confidence:** 2

**Summary:**

This paper considers a specific local differential privacy (LDP) framework, coined personalised LDP, where each user can set his/her individual privacy parameter. The authors are studying mean/sum estimation problems under this framework and provide two estimation protocols with theoretical foundations. Experiments illustrate the benefits of their approach.

**Questions:**

See above

**Ethical Concerns:**

["NO or VERY MINOR ethics concerns only"]

**Quality:**

2

**Strengths And Weaknesses:**

**Strengths**
* The paper is well-written, clear and sufficiently motivated.
* The problem is relevant to the community, and up to my knowledge, the proposed protocols and their theoretical insights are new.
* Related work is sufficient.

**Questions**
* Could the authors comment on the optimality of their error upper bounds for the two protocols?

---

> ### Author Rebuttal · Authors · 2025-07-27
>
> Reply to Question \#1:
>
> We agree that optimality is an important theoretical question to consider.  However, the answer is quite nuanced, depending on the precise notion of optimality.
>
> Worst-case optimality: Consider a particular user $u$, and the following instances the user may have: $(B,0,...,0), (0,B,...,0),..., (0,0,...,B)$.  Since LDP requires all of them to be $\Phi(u)$-indistinguishable, the user must injects a noise of scale $\Omega(\frac{B}{\sqrt{\Phi(u)}})$ on each coordinate (this can be formalized by standard lower bound arguments). Thus the worst-case error for the sum of all users' data is $\Omega(B\sqrt{d\cdot\sum_{i=1}^n \frac{1}{\Phi(u_i)} })$.  Our algorithms incur an error no more than this on any instance, so it is worst-case optimal.  However, this bound can also be achieved by a trivial mechanism that simply adds an $O(\frac{B}{\sqrt{\Phi(u)}})$ Gaussian noise on each coordinate, which is very large on typical instances.  So worst-case optimality is not very meaningful, either theoretically or practically.
>
> A much stronger notion of optimality is instance optimality.  However, for any instance $\mathbf{I}$, there always exists a trivial mechanism $M(\cdot) \equiv \mathrm{Sum}(\mathbf{I})$ that achieves 0 error on this instance (while incurs large errors on other instances).  This means that instance optimality is unattainable.
>
> Several prior works [13,18,A] has then introduced various relaxed notions of instance optimality, but they have not considered the personalized LDP model.  It would be an interesting direction to first extend these optimality notions to this model, and then see if our protocols are optimal with respect to any of them.
>
>
> [A] Dick, Travis, et al. "Subset-based instance optimality in private estimation." International Conference on Machine Learning. PMLR, 2023.

---

### Comment · Area_Chair_vDvj · 2025-08-05

Dear Reviewers,

Thank you for your valuable reviews. With the Reviewer-Author Discussions deadline approaching, please take a moment to read the authors' rebuttal and the other reviewers' feedback, and participate in the discussions and respond to the authors. Finally, be sure to complete the "Final Justification" text box and update your "Rating" as needed. Your contribution is greatly appreciated. I will flag irresponsible (final) reviews and/or any reviewers not participating in discussions.

Reviewers are expected to stay engaged in discussions, initiate them and respond to authors’ rebuttal, ask questions and listen to answers to help clarify remaining issues.

It is not OK to stay quiet.

It is not OK to leave discussions till the last moment.

If authors have resolved your (rebuttal) questions, do tell them so.

If authors have not resolved your (rebuttal) questions, do tell them so too.

Thanks.

AC

---

### Decision · Program_Chairs · 2025-09-17

**Decision:**

Accept (poster)

**Comment:**

Summary: This paper considers a specific local differential privacy (LDP) framework, coined personalised LDP, where each user can set his/her individual privacy parameter. The authors are studying mean/sum estimation problems under this framework and provide two estimation protocols with theoretical foundations. Experiments illustrate the benefits of their approach.

Strength: (1) This paper studies an important problem. (2) The method is well written and well motivated.

Weakness: (1) No experiments on real data and no comparison with previous work.

Based on the importance of the work and all reviewers agree to accept the paper. My suggestion is accept.